# Novel variants in *TECRL* leading to catecholaminergic polymorphic ventricular tachycardia

Douglas Jones[1,2], Jacob Hartung[1,2], Elizabeth Lasalle[1,3], Alejandro Borquez[1,3], Viridiana Murillo[4], Lucia Guidugli[4], Kiely N James[4], Stephen F Kingsmore[4], Nicole G Coufal[1,2,4]

**Pathogenic and likely pathogenic variants in the *TECRL* gene are known to be associated with recessive catecholaminergic polymorphic ventricular tachycardia 3, which can include prolonged QT intervals (MIM#614021). We report a case of cardiac arrest in a previously healthy adolescent male in the community. The patient was found to have a novel maternally inherited likely pathogenic variant in *TECRL* (c.915T>G [p.Tyr305Ter]) and an additional 19-kb duplication encompassing multiple exons of *TECRL* (chr4:65165944-65185287, dup [4q13.1]) not identified in the mother. Genetic results were revealed via rapid whole-genome sequencing, which allowed appropriate treatment and prognostication.**

## Case Presentation

A 12-yr-old male with autism presented to our institution after cardiac arrest; he was otherwise previously healthy with no familial cardiac history, on vacation. Per the history provided by the mother and bystanders, the patient was standing in two feet of ocean surf when a large wave unexpectedly knocked him over. He briefly resurfaced, stood, and then collapsed into the water. Lifeguards found the patient to be pulseless. An initial on-scene automated external defibrillator rhythm strip (Fig 1A) was concerning for an unusual polymorphic ventricular tachycardia (Fig 1A). Return of spontaneous circulation was achieved after three total 200 J defibrillation shocks were delivered and a single intraosseous 1 mg epinephrine was administered by Emergency Medical Technicians.

On arrival at the emergency department, the patient was in extremis with respiratory failure and depressed neurologic status (Glasgow Coma Scale 9) and therefore intubated and sedated. On admission to the pediatric intensive care unit, the patient was treated with an epinephrine infusion for refractory hypotension. An initial electrocardiogram (ECG) was notable for normal sinus rhythm with left axis deviation, a wide Q wave in lead II, and a prolonged QTc 468 ms (ref < 440 ms). Initial laboratory studies were most significant for mild troponin elevation (0.11 ng/ml, ref < 0.05 ng/ml), normal B-type natriuretic peptide (20 pg/ml, ref < 100 pg/ml), and elevated D-dimer (>20.00 µg/ml, ref < 0.50 µg/ml). An echocardiogram demonstrated normal coronary artery anatomy with qualitatively normal right and left ventricular size and function.

On hospital day (HD) 2, telemetry showed sinus tachycardia with occasional late premature ventricular complexes with pulsus alternans, the premature ventricular complex morphology possibly arising from high posterior RV outflow septal area (Fig 1B). At that time, an ECG was concerning for worsening prolonged QTc (QTc 568 ms, ref < 440 ms) and ST segment prolongation (Fig 1B). Because of the risk for arrhythmia in this setting, he was initiated on esmolol for rate control, and transitioned from epinephrine to norepinephrine and milrinone for ongoing hypotension and depressed cardiac function. After the addition of beta-blockade, no further ventricular ectopy was noted. A myocarditis/pericarditis panel (Quest Diagnostics) was negative for Echovirus, Coxsackie, Influenza A and B, and Chlamydophila pneumoniae antibodies. Serial troponin measurements peaked at 1.89 ng/ml (ref < 0.5 ng/ml) on HD2 and eventually normalized. He was off vasopressors by HD4 with normal cardiac function on echocardiography. A cardiac MRI was obtained, which was notable for normal left ventricular function, normal coronary origins, mildly diminished right ventricular function (46%), mild LV hypertrophy, and mild hypertrabeculations, which did not meet the criteria for LV non-compaction. The patient's QTc improved from the initial maximal prolongation of 568 ms but remained borderline prolonged between the ranges of 440 and 460 ms (ref < 440 ms). He was extubated on HD5.

## Technical Analysis and Methods

Parental consent was obtained for rapid whole-genome sequencing (rWGS), performed by the Rady Children's Institute for Genomic Medicine (RCIGM) as a clinically available and reportable study. Solo rWGS was performed at Rady Children's Institute for

---

[1]Rady Children's Hospital, San Diego, CA, USA   [2]Division of Pediatric Critical Care, Department of Pediatrics, University of California, San Diego, CA, USA   [3]Division of Cardiology, Department of Pediatrics, University of California, San Diego, CA, USA   [4]Rady Children's Institute for Genomic Medicine, San Diego, CA, USA

Correspondence: ncoufal@health.ucsd.edu

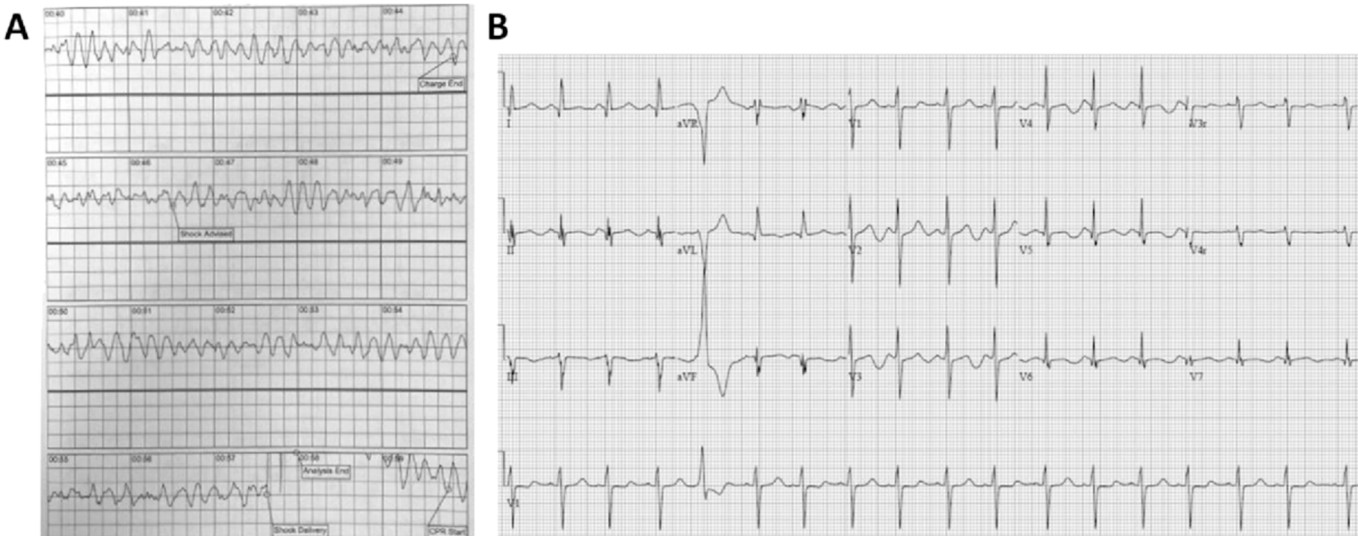

**Figure 1. Electrocardiograms.**
**(A, B)** On-scene automated external defibrillator rhythm strip consistent with polymorphic ventricular tachycardia; shock delivered at 00:57 (B) electrocardiogram on HD2 with a QTc interval of 568 ms (ref < 450 ms).

**Table 1. Clinical findings.**

| TECRL-associated ventricular tachycardia, catecholaminergic polymorphic 3, clinical features | Proband (II-1) | Mother (I-1) | Relevance/alternate explanation |
|---|---|---|---|
| HP:0006682, premature ventricular contractions (PVC) | Yes | Unknown | |
| HP:0001663, ventricular fibrillation | No | Unknown | Never noted on telemetry or initial rhythm strip |
| HP:0004751, paroxysmal ventricular tachycardia | Yes | No | |
| HP:0001695, cardiac arrest | Yes | No | |
| HP:0001962, palpitations | No | Unknown | |
| HP:0001657, prolonged QTc | Yes | Unknown | |
| Novel clinical features | | | |
| HP:0000729, autistic behavior | Yes | No | Assumed unrelated |
| HP:0031195, apical hypertrabeculation of the left ventricle | Yes | Unknown | Novel change. Not previously associated with TECRL variants |

Genomic Medicine as described (Owen et al, 2023). After DNA isolation from a whole blood sample, sequencing libraries were generated using the Illumina PCR-Free kit (Illumina) according to the manufacturer's instructions. Paired-end sequencing was performed on a NovaSeq 6000 instrument and S1 flow cell (Illumina). Read alignment to the reference human genome assembly GRCh37 and variant calling was performed using DRAGEN Bio-IT Platform v.3.10 (Illumina). Variants were annotated and analyzed using Fabric Enterprise version 6.16.14 (Fabric Genomics). The clinical phenotype is summarized in Table 1. The Human Phenotype Ontology (HPO) terms used during variant classification included Cardiac arrest (HP:0001695), Respiratory failure (HP:0002878), Prolonged QTc interval (HP:0005184), Torsade de pointes (HP:0001664), Pulmonary edema (HP:0100598), Aspiration pneumonia (HP:0011951), Lactic acidosis (HP:0003128), Abnormality of coagulation (HP:0001928), Seizure (HP:0001250), Autism (HP:0000717). Single-nucleotide variants (SNVs) and insertion/deletion events (indels) were filtered to retain variants with the allelic balance between 0.3 and 0.7,

prioritized by Phevor Gene Rank (Owen et al, 2023), and classified according to ACMG and AMP standards and guidelines (Richards et al, 2015). Copy-number variants were filtered to retain variants overlapping or have a boundary that lies within 1 kb of an exon in all coding genes and classified according to ACMG standards and guidelines (Kearney et al, 2011). The structural variant was orthogonally confirmed by multiplex ligation-dependent probe amplification in the proband and his mother. The father's sample was not available for analysis.

## Variant Classification

The patient was found to have a novel maternally inherited likely pathogenic variant in *TECRL* (c.915T>G [p.Tyr305Ter]) and an additional 19-kb duplication encompassing multiple exons of *TECRL* (chr4:65165944-65185287, dup [4q13.1]) not identified in the mother

**Table 2. Genomic variants identified.**

| Gene | Genomic location | HGVS cDNA | HGVS protein | Zygosity | Parent of origin | Variant classification |
|------|------------------|-----------|--------------|----------|------------------|------------------------|
| *TECRL* | 4:65147195 | c.915T>G | p.Tyr305Ter | Heterozygous | *Mother* | Likely pathogenic |
| *TECRL* | 4q13.1 | 4:65165944–65185287x3 | 19 KB duplication | Heterozygous | *Not maternal* | Likely pathogenic |

also classified as likely pathogenic (Table 2). These met ACMG PM4 criteria of likely pathogenicity by changing protein length because of premature termination. He did not have reportable variants in *KCNQ1*, *KCNH2*, *SCN5A*, *KCNE1*, or *KCNE2*, the genes that are most commonly associated with long QT syndrome. This combination of a heterozygous likely pathogenic nonsense variant and likely pathogenic duplication in the *TECRL* gene would explain the patient's clinical phenotype if the alterations were inherited in an autosomal recessive in trans fashion in the patient. *TECRL* (trans-2,3-enoyl-CoA reductase-like) is a gene expressed in myocardial cells within the endoplasmic reticulum that encodes a protein in the 5-alpha-reductase family. Likely pathogenic and pathogenic variants in *TECRL* are associated with catecholaminergic polymorphic ventricular tachycardia (CPVT), type 3 (Moscu-Gregor et al, 2020). Both variants are absent from the gnomAD population database.

Based on the rWGS result, and with the consultation of pediatric electrophysiology, both nadolol and flecainide were initiated. ECG and telemetry were monitored without significant ectopy, with both stable QRS and QT intervals. There was a risk assessment conversation between the family and care team regarding surgical intervention with an implantable cardioverter–defibrillator (ICD) or medical management paired with high-intensity monitoring via an implantable loop recorder (i.e., LINQ). After a multidisciplinary discussion, the patient was discharged to his out-of-state home on HD9 with medical management, a LINQ monitor, and close follow-up at his home institution. At this time, follow-up information is not available from this patient's home institution.

## Discussion

Channelopathies such as long QT syndrome (LQTS) and CPVT are associated with significant morbidity and mortality in pediatric patients (Offerhaus et al, 2020). These are the two most common channelopathies, with each containing multiple clinical subgroup variations. Both are inherited disorders with multiple pathogenic variants identified within each subgroup. LQTS is associated primarily with three genes: *KCNQ1* (encoding the *α* subunit of the IKs-conducting K+ channel), *KCNH2* (encoding the *α* subunit of the IKr-conducting K+ channel), and *SCN5A* (encoding the *α* subunit of the INa-conducting Na+ channel) (Wilde et al, 2022). These three genes are the source of most of the identified pathogenic variants in this syndrome. CPVT is primarily associated with variants in *CASQ2* (CPVT1, a calcium-binding protein) or *RYR2* (CPVT2, encoding a Ca2+ channel in the sarcoplasmic reticulum). Neither of these syndromes, however, is completely defined with possibly numerous additional genetic etiologies not yet established. An identified pathogenic variant in *TECRL* has been linked to an established clinical syndrome, which falls under the umbrella of both CPVT and

LQTS (MIM#614021). Pathogenic diplotypes in this gene are associated with elongated QT interval induced with stress states and normal baseline electrocardiograms. Devalla et al described three cases of an inherited autosomal recessive arrhythmia syndrome linked to the *TECRL* gene identified on whole-exome sequencing (Devalla et al, 2016). This case uniquely involves apical hyper-trabeculations and a heterogeneous presentation that includes aspects of both LQTS and CPVT. Structural heart defects have not previously been reported in TECRL patients. Unfortunately, a stress test was not available for review as the patient returned to care out of state.

Previous deleterious variants in patients with CPVT have been identified in the sixth exon of *TECRL* and the splice donor site of intron 3 resulting in an internal deletion (Devalla et al, 2016). A subsequent case report detailed compound heterozygosity in Arg196Gln and the c.918+3T > G splice-site altering variant resulting in CPVT (Xie et al, 2019). Our patient had two previously undescribed deleterious variants. The first was a maternally inherited nonsense likely pathogenic variant, c.915T>G (p.Tyr305Ter), in exon 10 leading to a premature stop codon and predicted to result in loss of normal protein function through either protein truncation or nonsense-mediated mRNA decay; however, this mechanism has not been confirmed by functional studies. The second variant was a non-maternal (a paternal sample was not available for analysis) intragenic duplication of approximately at 4q13.1, which encompasses exons 5–7 of the *TECRL* gene. Manual examination of the sequencing reads supported a tandem duplication, which is predicted to cause loss of normal protein function through either protein truncation or nonsense-mediated mRNA decay. Both variants are absent from the gnomAD population database.

*TECRL* is a 363–amino acid protein located on 4q13.1 and a part of the steroid 5-alpha-reductase family that is primarily expressed in the endoplasmic reticulum of myocardial and skeletal muscle cells. The exact molecular mechanism of TECRL-associated LQTS and CPVT3 is poorly understood, although analysis of intracellular calcium dynamics has shown that affected patients demonstrate prolonged action potentials, alterations in calcium handling, and a predisposition for delayed afterdepolarizing during catecholaminergic stimulation (Devalla et al, 2016). More recent research has also demonstrated endoplasmic reticulum membrane structures that can play a pivotal role in calcium signaling and energy metabolism (Hou et al, 2022).

Two recent case reports describe similar pathogenic variants in the *TECRL* gene leading to an eventual diagnosis of CPVT. First, Ebrahim et al (2023) report a case of an 11-yr-old boy with cardiac arrest, subsequent prolonged QTc after return of spontaneous circulation, and inducible unstable polymorphic VT with exercise in the electrophysiology laboratory. Whole-exome sequencing was notable for a previously described homozygous *TECRL* splice-site variant c.331+1G>A (Ebrahim et al, 2023). Secondly, two affected

Lebanese siblings experienced exercise-induced symptoms with an ECG notable for abnormal QT prolongation, abnormal T-wave morphology, and bidirectional ventricular ectopic beats during an exercise stress test. Genetic sequencing confirmed a novel *TECRL* variant, a homozygous c.742_758 del variant that resulted in a protein variant (p.Arg248Cysfs∗). In both cases, there was a strong family history of consanguinity and both patients eventually required dual therapy with both a beta-blocker and flecainide (Charafeddine et al, 2023).

The first-line therapeutic option for patients with CPVT involves non-selective beta-blockade (propranolol and nadolol) combined with exercise restriction. Nadolol, one of the longer acting beta-blockers, in high doses (1–2 mg/kg) is the preferred beta-blocker for prophylactic therapy and has been found to be more effective than selective beta-blockers (Leren et al, 2016). However, there remains an increased risk of a cardiac event in patients on beta-blockade, with one long-term follow-up study of 101 patients with CPVT demonstrating an 8-yr cardiac event rate of 27% for patients on beta-blockers and potentially fatal arrhythmic events in 11% of CPVT patients (Hayashi et al, 2009). This rate was significantly improved when compared to the 58% of patients not on a beta-blocker who experienced a cardiac event (Hayashi et al, 2009).

Flecainide, a class Ic antiarrhythmic drug, in addition to beta-blockers has been proven to be effective in reducing the ventricular arrhythmia burden. However, the exact mechanism remains controversial. Initial studies asserted that flecainide had a direct inhibitory action on RYR2 channels, which therefore limited sarcoplasmic reticulum calcium release and attenuated delayed afterdepolarization (Watanabe et al, 2009; Hilliard et al, 2010; Kryshtal et al, 2021). However, another study demonstrated that flecainide does not inhibit RYR2 channels directly. They instead hypothesized that flecainide's use-dependent block of Na+ may reduce the probability of RYR2 opening by reducing the level of cytosolic Ca2+ as a result of sodium–calcium exchanger–mediated Ca2+ efflux driven by reduced cytosolic Na+ (Sikkel et al, 2013; Bannister et al, 2015). Regardless of the mechanism, flecainide has demonstrated notable efficacy in the prevention of ventricular arrhythmias in patients with CPVT (Watanabe et al 2009, 2013; Devalla et al, 2016; Kannankeril et al, 2017; Kallas et al, 2021; Bergeman et al, 2023). A 2011 study demonstrated 76% of patients with CPVT on flecainide had either partial or complete suppression of exercise-induced ventricular arrhythmias. In addition, in a median follow-up of 20 mo, only 1 of 33 patients had an arrhythmic event (van der Werf et al, 2011).

ICD can be considered in children who demonstrate breakthrough arrhythmic events or sustained ventricular tachycardia or fibrillation despite antiarrhythmic treatment. ICD implantation, however, comes with its own risk of device-related complications and has the potential risk for proarrhythmic effects because of both appropriate and inappropriate discharges leading to potentially life-threatening arrhythmias (Mohamed et al, 2006; Pizzale et al, 2008). Finally, there have been studies published that report significant results for the use of left cardiac sympathetic denervation (LCSD) on cardiac events in patients with refractory disease to pharmacological treatment (Wilde et al, 2008). The strategy of LCSD focuses on interfering with catecholamine-induced pathways that activate the release of calcium from the sarcoplasmic reticulum and therefore increasing the threshold for ventricular fibrillation.

The prognosis of patients with CPVT depends on early identification and adequate treatment. There are limited data on long-term prognosis for TECRL-associated CPVT3 specifically. In general, CPVT left untreated is a severe disease with high mortality with 30% of affected patients experiencing at least one cardiac arrest and up to 80% with one or more syncopal spells (Priori et al, 2002). As stated previously, even those started on appropriate beta-blocker therapy continue to be at increased risk of serious arrhythmias and require further pharmacological treatment with flecainide. Even with optimized medical treatment, there is a subset of patients who continue to have arrhythmic events and will require ICD or further surgical intervention with LCSD. Altogether, this case identified novel variants in *TECRL* associated with CPVT leading to sudden cardiac arrest and extended the clinical features associated with *TECRL* variants.

Overall, we describe two novel variants in the *TECRL* gene on chromosome 4q13.1 associated with ventricular tachycardia and cardiac arrest in a previously asymptomatic patient. His presentation, intra-arrest electrocardiographic findings, and the patient's young age raised suspicion of CPVT, type 3. The rWGS findings were consistent with the clinical suspicion of CPVT. This finding impacted treatment decisions, including choice of antiarrhythmic medications and prognosis, and extends on the clinical features associated with TECRL to include apical hypertrabeculations.

## Data Availability

ClinVar accession number VCV001766009.3, Variation ID: 1766009, and Identifiers NM_001010874.5 (TECRL):c.915T>G (p.Tyr305Ter) are available at https://www.ncbi.nlm.nih.gov/clinvar/variation/1766009/.

### Ethics statement

Informed and signed consent forms were obtained for all sequenced individuals in this study. The project is approved by the Institutional Review Board of the University of California at San Diego and has received non-significant risk status in a pre-Investigational Device Exemption submission to the Food and Drug Administration.

## Acknowledgements

This study was supported by Rady Children's Hospital, and National Institute of Child Health and Human Development and National Human Genome Research Institute grant U19HD077693.

### Author Contributions

D Jones: formal analysis, investigation, visualization, and writing—original draft, review, and editing.
J Hartung: formal analysis, visualization, and writing—original draft, review, and editing.

E Lasalle: investigation, methodology, and writing—original draft.
A Borquez: investigation, visualization, methodology, and writing—original draft.
V Murillo: investigation, methodology, and writing—original draft.
L Guidugli: investigation, methodology, and writing—original draft.
KN James: investigation, methodology, and writing—original draft.
SF Kingsmore: resources, project administration, and writing—original draft.
NG Coufal: conceptualization, resources, data curation, formal analysis, methodology, and writing—original draft, review, and editing.

## Conflict of Interest Statement

The authors declare that they have no conflict of interest.

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
